# Can a Two-Dose Influenza Vaccine Regimen Better Protect Older Adults? An Agent-Based Modeling Study

**DOI:** 10.3390/vaccines10111799

**Published:** 2022-10-26

**Authors:** Katherine V. Williams, Mary G. Krauland, Lee H. Harrison, John V. Williams, Mark S. Roberts, Richard K. Zimmerman

**Affiliations:** 1Department of Family Medicine, School of Medicine, University of Pittsburgh, Schenley Place, 5th Floor, Suite 520, Pittsburgh, PA 15260, USA; 2Department of Health Policy and Management, School of Public Health, University of Pittsburgh, Pittsburgh, PA 15261, USA; 3Public Health Dynamics Laboratory, School of Public Health, University of Pittsburgh, Pittsburgh, PA 15261, USA; 4Center for Genomic Epidemiology, Division of Infectious Diseases, School of Medicine, University of Pittsburgh, Pittsburgh, PA 15261, USA; 5Department of Pediatrics, Division of Pediatric Infectious Disease, School of Medicine, University of Pittsburgh, Pittsburgh, PA 15224, USA

**Keywords:** influenza, vaccine, adult

## Abstract

Older adults (age ≥ 65) are at high risk of influenza morbidity and mortality. This study evaluated the impact of a hypothetical two-dose influenza vaccine regimen per season to reduce symptomatic flu cases by providing preseason (first dose) and mid-season (second dose) protection to offset waning vaccine effectiveness (VE). The Framework for Reconstructing Epidemiological Dynamics (FRED), an agent-based modeling platform, was used to compare typical one-dose vaccination to a two-dose vaccination strategy. Primary models incorporated waning VE of 10% per month and varied influenza season timing (December through March) to estimate cases and hospitalizations in older adults. Additional scenarios modeled reductions in uptake and VE of the second dose, and overall waning. In seasons with later peaks, two vaccine doses had the largest potential to reduce cases (14.4% with February peak, 18.7% with March peak) and hospitalizations (13.1% with February peak, 16.8% with March peak). Reductions in cases and hospitalizations still resulted but decreased when 30% of individuals failed to receive a second dose, second dose VE was reduced, or overall waning was reduced to 7% per month. Agent-based modeling indicates that two influenza vaccine doses could decrease cases and hospitalizations in older individuals. The highest impact occurred in the more frequently observed late-peak seasons. The beneficial impact of the two-dose regimen persisted despite model scenarios of reduced uptake of the second dose, decreased VE of the second dose, or overall VE waning.

## 1. Introduction

Despite 60–70% influenza vaccine uptake among older adults (≥65 years) [1], the risk of morbidity and mortality from influenza is high. An estimated 50–70% of influenza-related hospitalizations and 70–85% of influenza-related deaths in the United States in recent years occurred among older adults [2]. One reason may be waning immunity during the influenza season.

The influenza season is typically considered to occur during October through March in the Northern Hemisphere, with a peak in cases most commonly observed in February [3]. Waning immunity over this period could leave vaccinated older adults with inadequate protection against influenza infection. For example, in the 2015–2016 through the 2018–2019 seasons, vaccine effectiveness (VE) approached zero by 5–6 months after vaccination, when adjusted for factors including season, age, race, and calendar time of illness onset [4]. In hospitalized patients, VE waning can be as high as 10–11% per month in older adults compared to 8–9% in all adults (≥18 years) [4].

Influenza vaccinations typically occur between September and November [5]. Accounting for a two-week period after vaccination to reach peak vaccine efficacy [6], peak protection would occur between mid-September to mid-December. Ideally, peak VE would occur during the peak risk period for infection, yet with the current one-dose preseason vaccination recommendations, this period of peak protection could be as much as 2 months before the peak of the influenza season in typical seasons. Furthermore, with the variability of the onset and peak of influenza seasons from year to year, timing vaccination with a single dose to optimize VE and offset waning is challenging and may leave a substantial portion of older adults relatively unprotected later in the season.

Two influenza vaccine doses separated by 3 months within one season for older adults are a potential solution but have not been tested. Agent-based modeling that incorporates individual level data is ideal for comparing different vaccine dosing schedules, while varying influenza season onsets and peaks and waning VE. We used the Framework for Reconstructing Epidemiological Dynamics (FRED) platform to simulate the effects of the current one-dose vs. a hypothetical two-dose influenza vaccination schedule on influenza cases and hospitalizations in older adults.

## 2. Materials and Methods

FRED is an agent-based (agent defined as an individual person) modeling platform that uses census-based synthetic populations with household demographics, incomes, and locations that are statistically realistic at the US census block group level. In FRED, agents spread infectious conditions through interactions in modeled locations (schools, workplaces, neighborhoods, and households). FRED models are composed of conditions (e.g., influenza) that include disease definitions as a series of health states that individuals can progress between. For example, an individual within the influenza condition can transition to other states within that condition (e.g., those in the susceptible state may transition to the exposed state) based on probabilities and time periods for evaluation of state changes. FRED has been used to model influenza and other diseases and conditions and has been described in detail [7,8,9,10]. The models and parameters used in this study are described in more detail in Appendix A.

### 2.1. Model Inputs

For this study, FRED tracked a population of 1.2 million people created from the 2010 Allegheny County Pennsylvania census residing in urban or suburban areas. One- and two-dose vaccination strategies were modeled over a single season starting 15 August and ending 31 May, across a range of influenza seasons. Input parameters and simulation descriptions are included in Appendix A. The influenza season was started by inserting 50 cases into the population on a range of seeding dates (September through December), with an effective reproductive rate of ~1.5, resulting in peak cases from December to March (Epidemiologic curve by seeding date, Figure 1). The simulation included a single strain of influenza, similar to type A in terms of burden. 

To reproduce the seasonal timing of influenza (December through March in this study) [11], all models included a change in the transmissibility parameter that depends on how far the current day is from the winter solstice [12]. This seasonal transmissibility parameter is intended to account for factors that may influence transmissibility across a range of seasons due to variability in factors such as temperature, humidity, and changes in contact rates [13]. Results with seasonality were compared with the single dose baseline model with vaccinations beginning 1 September without seasonality. The seasonality induced reduction in cases reverts when seasonality is removed.

### 2.2. Model Structure

This study used a modified Susceptible-Exposed-Infectious-Recovered (SEIR) model (Appendix A) [7] with the addition of varying susceptibility to infection by age. Individuals over age 75 were assigned higher base susceptibility to influenza infection than younger age groups, i.e., by 1.5 times for age 75–84 years and 2.5 times for age ≥85 years [14]. Susceptibility was reduced to 0 after infection.

Hospitalization estimates for symptomatic individuals were based on national CDC estimates of 9% hospitalization rates in ages 65 and up [14] with higher rates in the age ranges of 75–84 years and ≥85 years [15,16]. Hospitalization estimates for cases of symptomatic illness were 6% for age 65–74 years, 12% for age 75–84 years, and 30% for age ≥85 years, representing a 2- and 5-fold increase, respectively, for these age strata compared to age 65–74 [15,16].

For flu vaccination, all age groups were vaccinated as per CDC 2019 reporting, including 68.7% of those ≥65 years (Appendix A) [5]. Agents were randomly chosen for application of vaccination. Baseline VE was set at 40% for all ages for both doses based on mean reported VE in 2004–05 to 2019–20 seasons (mean 40%, range 10% to 60%) [17]. Waning of VE was set at 7% per month for age <65 years and 10% per month for age ≥65 years [4,6]. Modeling steps are described in Appendix A and a representation of VE as modeled is shown in Appendix A. Since FRED models are stochastic, each model was run 100 times to produce mean values and 95% confidence intervals for cases and hospitalizations.

### 2.3. Vaccination Simulation Base Model Scenarios

The base model simulated the typical one dose annual influenza vaccination pattern [5] relative to a range of seasonal influenza case peaks (Table 1). In every model, all individuals received one influenza vaccine dose beginning 1 September over a mean of 45 days with a standard deviation of 14 days (Appendix A). In the primary comparison model, individuals ≥65 years received a second dose 90 days after the first.

### 2.4. Additional Scenario Analyses

Additional scenario analyses were performed to (1) estimate the impact of reduced vaccine uptake for the second dose by 30% (i.e., only 70% of vaccinees age ≥65 receive a second dose); (2) reduced VE for the second dose from 40% to 30%; and (3) waning of vaccine-induced immunity in age ≥65 from 10% to 7%. The VE of two influenza vaccines administered 90 days apart within a season is unknown, so a conservative estimate for VE for the second dose of either no reduction in VE (base model) or reduced VE as may occur if circulating viral antigens change over time due to antigenic drift. Reduced waning (7%) was included based on studies that showed a lower degree of waning in outpatient populations that was not specific to age but may vary based on influenza type [6].

### 2.5. Comparison to Prior Models

Prior studies showed a reduction in influenza cases when a single influenza vaccine dose was delayed until 1 October [18,19]. For comparison to the single and two dose models, additional simulations across seasons that included one vaccine dose with later administration in age ≥65, beginning 1 October over a mean of 30 days with a standard deviation of 9 days (October–November), were performed.

The University of Pittsburgh Institutional Review Board determined that this study was not human subjects’ research.

## 3. Results

### 3.1. Base Model

The single dose base model for age ≥65 with similar vaccination uptake rates to those reported by the CDC and VE of 40% resulted in a mean of 1985 to 3144 cases per 100,000, depending on the timing of peak influenza cases (Table 2). Later seasons had fewer cases due the incorporation of a seasonality parameter into all models. Hospitalizations ranged from 194 to 313 per 100,000 and increased across each age stratum within a season (Table 3), based on model parameters set to approximate a 9% overall hospitalization rate of symptomatic cases for age ≥65, with higher rates of hospitalizations in ages 75–85 and ≥85 and up.

When a second vaccine dose 90 days after the first was added to the model, cases declined (Table 2). When compared by season, a second vaccine dose decreased total cases in ages ≥65 by 1.6% to 18.7% (Figure 2A), with the greatest impact observed with the latest seasonal influenza case peaks and lesser impact with earlier season peaks. Correspondingly, a decrease in cases resulted in a decrease in hospitalizations (Table 3) ranging from 2.2% to16.8% depending on season peak (Figure 2A).

Despite increased susceptibility and hospitalization rates in ages 75–84 years and ≥85 years, compared to ages 65–74 years, the percentage reduction in cases and hospitalizations due to the addition of the second influenza vaccine was highest in ages 65–74 (Appendix A). For example, for a February flu season peak, cases decreased by 15.8% and hospitalizations decreased by 15.0% among those 65–74 years, but the decreases were lower in ages ≥85 with a decrease in cases of 10.7% and a decrease in hospitalizations of 11.0%.

### 3.2. Scenario Analyses

Scenario analyses across all seasons studied examined how changes in model parameters could influence the impact of the second vaccine dose. In general, the findings of the base scenario of improved protection by a two-dose regimen were upheld in scenario analyses. For example, when uptake of the second vaccine was reduced to 70% of ≥65-year-old vaccinees receiving a second dose, cases decreased by a maximum of 11.9% and hospitalizations decreased by a maximum of 11.4% depending on the season (Figure 2B). In a model with 40% VE for dose one and 30% VE for dose two, cases were reduced by 9.8% and hospitalizations by 8.5% in a peak influenza season of February and cases were reduced cases by 12.8% and hospitalizations by 12.1% in a peak influenza season of March, compared to a single vaccine dose (Figure 2C). Reduced waning of VE also showed reductions as high as 12.8% for cases and 12.1% for hospitalizations (Figure 2D). Scenario analyses by age showed a pattern similar to the primary one- vs. two-dose comparison, with the greatest percentage reductions in age 65–74-year stratum among those ≥65 years (Appendix A).

We also modeled scenarios in which a one-dose influenza strategy was delayed until 1 October, to extend protection from a single vaccine dose further into the influenza season. This compressed strategy in which vaccination began 1 October over a mean of 30 days, was compared with the base model in this study in which vaccination began earlier (1 September) over a mean of 45 days. When a single vaccine was administered beginning 1 October, the maximum case reduction occurred when cases peaked in February. Compared to a single dose administered beginning 1 September, cases were reduced by a maximum of 4.1% with the delayed 1 October strategy, but the magnitude was less than the maximum 14.4% reduction in cases for a two-dose vs. one-dose schedule (Appendix A, last column, February peak). When a single vaccine was administered beginning 1 October, the maximum case reduction occurred when cases peaked in February. 

## 4. Discussion

Agent-based modeling was used to predict the impact of a two-dose influenza vaccine strategy to address waning of influenza VE over the six-month season. Results indicate that in general, two influenza vaccine doses could have a substantial impact in reducing cases and hospitalizations among older individuals, especially in seasons that peak in February and March. This added protection could be valuable in most seasons, given that influenza peaked in these months in 60% of the past 40 years [3]. CDC estimates that during a season of moderate severity such as 2019, 170,000 hospitalizations and 1 million medical visits due to influenza occurred in people ≥65 [20]. We estimated that a two-dose influenza vaccine regimen would result in a 15% reduction in influenza disease, which translates to 282,000 fewer symptomatic illnesses, >150,000 fewer medical visits and >25,000 fewer hospitalizations in this age group.

A two-dose vaccination strategy may be a way to counteract waning VE associated with increased hospitalizations [4]. The first dose distribution schedule used in the model followed the typical pattern in which the population receives influenza vaccines in September through November [21]. A ninety-day interval between doses was chosen for the sake of consistent coverage over a 6-month time span. In practice, a second dose could be scheduled 90 days later, or offered during a particular month based on a general recommendation (e.g., return for second dose in January). Current CDC guidelines recommend a single influenza vaccination be administered ideally by the end of October, avoiding July and August when possible, and that vaccine continue to be offered as long as influenza circulates locally [22].

Healthcare provider and patient acceptance of two influenza vaccine doses is unknown. In our primary comparison, every agent who received a first dose received a second dose. In practice, uptake for a second influenza vaccine may be lower. Even accounting for 30% of individuals not receiving a second dose, fewer cases and hospitalizations occurred. While the biggest impact of the two-dose schedule was observed in seasons that peaked in February and March (the farthest away from initial vaccination), the effect of two doses was consistent across seasons that peaked sooner.

We assumed that VE was identical for both doses in our primary model; however, the VE of a second dose administered 90 days after the first is unknown. Second non-influenza vaccine doses generally provide a booster effect within a short time interval; however, influenza is the only vaccine with annually updated components due to ongoing viral antigenic drift. Repeated influenza vaccination response is a complex topic and repeat vaccination does not typically occur within a season. Repeat vaccination across seasons may increase VE because the previous vaccine’s effectiveness is waning or decrease VE because of interference from antibodies produced in response to the previous year’s vaccine [23,24]. We chose a conservative estimate for the second influenza vaccine dose of either no reduction in VE or a slight reduction in VE. Additional clinical studies may enhance the understanding of the immunologic and clinical effects of two doses of influenza vaccine within an influenza season.

Waning immunity plays a significant role in long-term VE and may differ not only by age [4] but also influenza strain. For example, during 2011–2012 through 2014–2015 for ages 18 and up seeking outpatient care for acute respiratory illness, VE waned from 6–11% per month, comparable to the parameters of 7–10% for our models but depended on the influenza strain rather than age; waning was 7% per month for influenza A(H3N2) and 6–11% per month for influenza A(H1N1)pdm09 [6]. Even in models that assumed a lower level of waning of VE, the value of a second, mid-season dose of influenza vaccine was evident.

Another proposed strategy to address optimizing vaccine uptake with declining VE is to postpone influenza vaccination until October for older adults. Both a Markov model to estimate influenza cases [19] and a health state transition model to estimate hospitalizations [18] showed that these altered vaccination schedules could reduce influenza cases and/or hospitalizations. These benefits were offset by decreased uptake of influenza vaccine by those who would normally be vaccinated in August or September. If as few as 5–14% of individuals who would have been vaccinated in August or September failed to get vaccinated at all, the benefits of the delayed timing of vaccination were lost [18,19]. In our model, postponing vaccination until 1 October also reduced cases but the magnitude was lower than for the two-dose strategy. Allowing the first dose to occur by choice as early as September as in our model would likely reduce the chance of an overall missed vaccination opportunity, and beneficial effects persisted even when 30% of individuals did not receive a second dose.

Logistics of a two-dose vaccine schedule should be considered relative to variability in benefit within an age group of older adults. Higher susceptibility and hospitalization rates with increasing age were incorporated into the present model, and all age subgroups ≥65 benefited from two doses, suggesting two doses need not be restricted to a particular age subgroup to achieve results. The expense of a second vaccine dose could potentially be offset by the reduction in medical visits and hospitalizations but would require a separate cost-effectiveness analysis. 

A strength of this study is the application of FRED as an agent-based model to use a statistically realistic population to examine transmission by demographic structure (household, school, work, and community), stratification of population by age, and seasonal vaccination strategies across time. Multiple reports indicate FRED’s ability to represent epidemics [7,8,9,10].

Our model included a low number of cases, representing an influenza season of mild to moderate intensity [25]. We made adjustments to susceptibility and designation as symptomatic to attain cases comparable to CDC estimates. Recent mitigation efforts for respiratory viruses due to COVID-19 may result in lower numbers of influenza cases, potentially reducing the overall impact of this model for influenza vaccination in the current the COVID-19 era. Cases may also vary based on a variety of factors, including the circulating influenza strain, timing of the season, and prior immunity and we only modeled a single influenza strain. Modeling other strains may produce dissimilar results. Future directions could address cost effectiveness analyses.

Model parameters were chosen based on VE, seasonal patterns, and CDC hospitalization and case estimations over the past decade, but the model results presented are not specific to vaccine type, influenza strain, or year. While several different influenza vaccines are available, the model represents a generalized vaccine, given to the entire population, and assumes the same efficacy for all age groups, which may be an oversimplification. Higher dose and adjuvated influenza vaccines in older adults are now recommended to boost immunity [26], although VE also varies annually and the effects of these vaccines on waning are unknown. Because the peak of influenza season cannot be predicted, analyses should be conducted to determine if an annual two-dose regimen for older adults would be cost-effective, feasible and applicable to all vaccine strains.

## 5. Conclusions

In a mild to moderate, late influenza season, a two-dose influenza vaccine regimen with one preseason and one mid-season dose has the potential to decrease cases and hospitalizations in older adults who are among the most vulnerable to the adverse consequences of influenza. The benefits of two doses persisted even if all individuals did not receive the second dose, in scenarios of decreased VE of the second dose, and less waning than is believed to occur.

## Figures and Tables

**Figure 1 vaccines-10-01799-f001:**
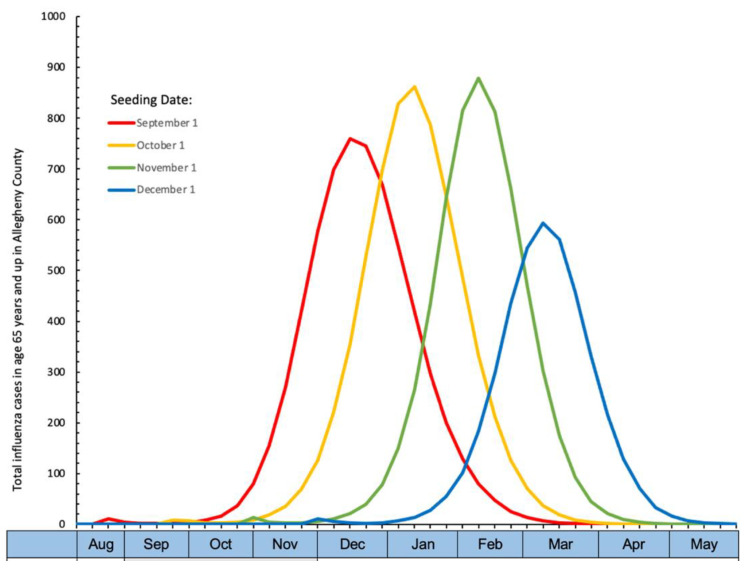
Epidemiologic curve by seeding date. Season peaks by seeding date with one dose of influenza vaccine starting 1 September over a mean of 45 days following a normal distribution. Variations in total cases by season peak are due to incorporation of the seasonal transmissibility parameter in the model to simulate seasonal outbreaks of the same influenza strain that peak at different times.

**Figure 2 vaccines-10-01799-f002:**
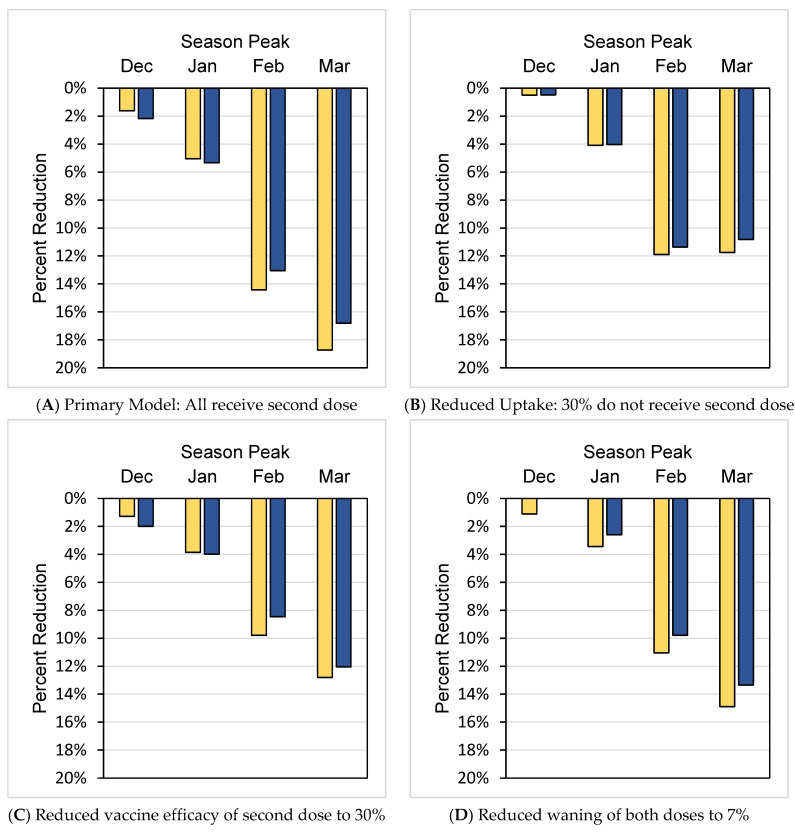
Percent reduction in cases and hospitalizations age 65 and up per 100,000 population with two vaccine doses by season peak. Yellow bars represent cases and blue bars represent hospitalizations.

**Table 1 vaccines-10-01799-t001:** Study Design: Two dose vaccination strategies. 90-day: Single dose: Doses occur over 90 days beginning 1 September following a normal distribution. Two Dose: Doses occur over 90 days beginning 1 September following a normal distribution. The second dose occurs 90 days after the first. Full vaccine effectiveness begins two weeks after dose 1, is sustained for two weeks, then wanes daily one month after dose 1. Waning from dose 1 stops for two weeks after dose 2. Full vaccine effectiveness begins again two weeks after dose 2, then wanes daily one month after dose 2.

Model	Aug	Sep	Oct	Nov	Dec	Jan	Feb	Mar	Apr	May
					Range of Season Peaks					
Single Dose		Dose 1, normal distribution starts 1 Sep								
Two Dose		Dose 1, normal distribution starts 1 Sep			Dose 2, 90 days after dose 1					

**Table 2 vaccines-10-01799-t002:** Symptomatic influenza cases per 100,000 population by peak month and by age group in ages 65 and up *.

Season Peak	Age 65–74 Years	Age 75–84 Years	Age 85 and up	Overall: Age 65 and up
December				
One dose	4006 (3787, 4259)	2394 (2251, 2511)	1568 (1398, 1729)	3010 (2860, 3156)
Two doses	3936 (3745, 3931)	2375 (2215, 2508)	1542 (1381, 1717)	2962 (2849, 3116)
January				
One dose	4221 (4024, 4416)	2477 (2337, 2689)	1585 (1449, 1789)	3144 (3021, 3293)
Two doses	3985 (3791, 4191)	2373 (2193, 2548)	1523 (1398, 1701)	2985 (2836, 3108)
February				
One dose	3932 (3629, 4190)	2215 (1956, 2326)	1391 (1220, 1608)	2880 (2644, 3094)
Two doses	3312 (2850, 3670)	1938 (1648, 2165)	1242 (1033, 1422)	2465 (2104, 2735)
March				
One dose	2751 (2194, 3305)	1493 (1099, 1804)	916 (686, 1132)	1985 (1577, 2382)
Two doses	2190 (1691, 2617)	1249 (942, 1519)	793 (587, 1024)	1613 (1242, 1931)

* Mean and 95% confidence interval over 100 simulations.

**Table 3 vaccines-10-01799-t003:** Influenza hospitalizations per 100,000 population by peak month and by age group in age 65 and up *.

Season Peak	Age 65–74 Years	Age 75–84 Years	Age 85 and up	Overall: Age 65 and up
December				
One dose	247 (212, 283)	297 (261, 334)	473 (393, 554)	302 (296, 311)
Two doses	240 (209, 268)	291 (254, 335)	466 (375, 545)	295 (270, 31,937)
January				
One dose	260 (225, 299)	307 (270, 354)	480 (396, 572)	313 (285, 341)
Two doses	244 (210, 278)	291 (256, 330)	457 (377, 555)	296 (273, 323)
February				
One dose	243 (200, 279)	277 (230, 327)	416 (342, 500)	283 (249, 315)
Two doses	207 (169, 254)	243 (190, 290)	370 (275, 443)	246 (198, 283)
March				
One dose	171 (126, 216)	186 (133, 250)	277 (183, 381)	194 (145, 246)
Two doses	137 (95, 169)	159 (110, 214)	237 (165, 311)	161 (118, 196)

* Mean and 95% confidence interval over 100 simulations.

## Data Availability

Not applicable.

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
