# Peer review of "Can a Two-Dose Influenza Vaccine Regimen Better Protect Older Adults? An Agent-Based Modeling Study"

_vaccines, 2022, doi:10.3390/vaccines10111799_

Round 1

Reviewer 1 Report

Comments:

This is an interesting study in which authors used agent-based modelling to evaluate the protective effect of a two-dose influenza vaccine regimen in older adults. The work is clearly presented, and the conclusions are logic. The authors described clearly the methods used, which is important for this work since the readers of this journal are probably 99% related to biological sciences.

Minor comments:

-        Authors should be careful with unnecessary spaces between sentences.

-        The journal template should show the current year and not 2021.

-        The quality of figures must be improved. For example, delete black and grey rectangles.

-        Please incorporate the figure showing the epidemiologic curve by seeding date in the main article and not as a supplementary figure.

Author Response

Authors should be careful with unnecessary spaces between sentences.-We have carefully reviewed to confirm two spaces (not three) between sentences.

The journal template should show the current year and not 2021.-This has been updated to 2022.

The quality of figures must be improved. For example, delete black and grey rectangles. Black and gray rectangles in current Figure 1 were deleted.  Black and gray rectangles in current Figure A3 were color coded and their representation clarified in the figure legend.

Please incorporate the figure showing the epidemiologic curve by seeding date in the main article and not as a supplementary figure.  This figure was moved to the main article and Supplementary figure numbers were updated.

Reviewer 2 Report

Williams et al. evaluated the efficacy of a hypothetical two-dose influenza vaccine regimen using an agent-based modeling platform. Although it is a modeling study with hypothetical assumptions, the conclusion of the manuscript is supported by the data presented.

1)    In the discussion section, “but the model results presented are not specific to vaccine type, influenza strain, or year” is described. Currently, several different influenza vaccines are available – the regular and high dose inactivated vaccines with or without adjuvant, although the high dose and adjuvanted vaccines started being recommended only recently. The live attenuated vaccine is also on the market, although it is not approved for > 50. To avoid any confusion, it is desirable to define what kind of influenza vaccine was administered to the data population used for this modeling study (i.e., 2010 Allegheny County Pennsylvania census).

2)    In the discussion section, “We assumed that VE was identical for both doses in our primary model; however, the VE of a second dose is unknown.” is described. Generally, the second dose provides higher protection by a booster if the booster is given within a short interval. However, the authors appear to assume that the same degree of protection is obtained with the second dose as with the first dose. This assumption is critical for the modeling and should be clarified in the method and results sections.

Author Response

Response to comment 1.  We acknowledge the reviewer's comment on the range of vaccines available.  On page 9, we added "While several different influenza vaccines are available, the model represents a generalized vaccine, given to the entire population, and assumes the same efficacy for all age groups, which may be an over simplification."  We also acknowledge in the original manuscript that "Higher dose and adjuvated influenza vaccines in older adults are now recommended to boost immunity, although VE also varies annually and the effects of the vaccines on waning are unknown."

Response to comment 2.  The reviewer's point is well taken, and we have expanded this explanation in section 2.4 of the Methods and the second full paragraph of the discussion on page 8.  

In summary, second non-influenza vaccine doses may often provide a booster effect within a short time interval, however influenza is the only vaccine with annually updated components due to ongoing antigenic drift that may even occur within a season.  

The response to multiple influenza vaccines is a complex topic, and no data on the vaccine effectiveness of two influenza doses within a season in older adults could be identified.  We chose a conservative estimate for the second influenza vaccine dose of either no reduction in VE or a slight reduction in VE.  Additional clinical studies may enhance the understanding of the immunologic and clinical effects of two doses of influenza vaccine in adults within an influenza season.